# Atomic-scale mapping of hydrophobic layers on graphene and few-layer MoS$_2$ and WSe$_2$ in water

Manuel R. Uhlig[1], Daniel Martin-Jimenez[1] & Ricardo Garcia [1]

The structure and the role of the interfacial water in mediating the interactions of extended hydrophobic surfaces are not well understood. Two-dimensional materials provide a variety of large and atomically flat hydrophobic surfaces to facilitate our understanding of hydrophobic interactions. The angstrom resolution capabilities of three-dimensional AFM are exploited to image the interfacial water organization on graphene, few-layer MoS$_2$ and few-layer WSe$_2$. Those interfaces are characterized by the existence of a 2 nm thick region above the solid surface where the liquid density oscillates. The distances between adjacent layers for graphene, few-layer MoS$_2$ and WSe$_2$ are ~0.50 nm. This value is larger than the one predicted and measured for water density oscillations (~0.30 nm). The experiments indicate that on extended hydrophobic surfaces water molecules are expelled from the vicinity of the surface and replaced by several molecular-size hydrophobic layers.

---

[1] Materials Science Factory, Instituto de Ciencia de Materiales de Madrid (ICMM), CSIC, 28049 Madrid, Spain. Correspondence and requests for materials should be addressed to R.G. (email: r.garcia@csic.es)

The interfacial water structure plays a key role in a large variety of interactions and processes such as wetting[1], molecular recognition[2,3], and patterning[4,5]. A variety of experimental and simulations methods has led to significant advances in our understanding of how water molecules interact with surfaces[3,6–9]. However, the structure and the role of the interfacial water in mediating the interactions and properties of surfaces immersed in aqueous environments is not well understood. This is due to the lack of suitable experimental tools and theoretical methods to explore non-ideal solid–liquid interfaces with atomic resolution.

Two-dimensional (2D) and few-layer materials provide a variety of extended atomically flat surfaces with different long-range (van der Waals) and hydrophobic interactions. In addition, 2D materials have attracted considerable interest because of the electronic, optical, thermal, and mechanical properties that arise from their atomic-size thickness[10–13]. Among other applications, those materials have been proposed as active elements in the development of chemical and biological sensors[14–16]. The use of 2D materials as active components of chemical, biological, and medical devices involves their direct interaction with water molecules. Likewise, the interfacial water structure should influence the interactions of 2D materials with other materials, solvents, ions, and small biomolecules.

In this context, the understanding of the interaction of water molecules with 2D materials is relevant for two reasons. First, those materials provide a variety of extended and atomically flat surfaces to investigate with atomic resolution hydrophobic interactions. Second, the understanding the interfacial water structure on 2D materials could lead to more efficient 2D materials-based sensors.

Macroscopic measurements such as water contact angle measurements show that the hydrophobicity of graphene, few-layer $MoS_2$, and $WSe_2$ increases with time due to the adsorption of airborne hydrocarbon contaminants[17–19]. Those results are supported by theoretical calculations[20]. At the sub-micrometer scale, AFM-based methods have been applied to study the wettability and the condensation of water vapor on graphene and few-layer $MoS_2$[14,21,22]. Those studies were either focused on the adsorption of water from air under different conditions (temperature and/or relative humidity)[22] and/or the influence of those processes on the electronic or structural properties of the 2D materials[21].

The graphene–water interface has been studied with sub-nanometer spatial resolution by AFM[23], molecular dynamics simulations (MD)[15,24], and X-ray reflectivity measurements[25,26]. Molecular resolution AFM experiments performed on graphene[23] have shown the formation of two solvation layers separated by 0.52 nm. This separation was significantly larger than the one found on mica in the same study (0.23 nm). MD simulations performed on graphene have shown the formation of a single hydration layer separated from the graphene by 0.29 nm[15].

We have developed a three-dimensional (3D)-AFM microscope[27] operated in the amplitude modulation regime[28] to expand the $z$-depth range of 3D-AFM[8]. Here, the 3D-AFM is applied to characterize with angstrom resolution the three-dimensional structure of the water near the surface of graphene and few-layer $MoS_2$ and $WSe_2$. Specifically, we report the observation of an oscillating structure characterized by the presence of up to three solvation layers within the last 2 nm of the liquid. The interfacial liquid features on those 2D materials are more pronounced than the ones observed on an archetypical hydrophilic surface (mica). The distance between the 1st solvation layer and the 2D surface shows some dependence on the material. It ranges for graphene, $MoS_2$ and $WSe_2$, respectively, 0.36, 0.31, and 0.35 nm. The distance measured between the first two adjacent interfacial layers is quite similar for the three

interfaces. The distances are for graphene, $MoS_2$ and $WSe_2$, respectively, 0.50, 0.50, and 0.48 nm. Those values are substantially larger than the ones measured by the same instrument on a mica surface which are, respectively, 0.18 and 0.34 nm. In fact, MD simulations performed for three tip-surface interfaces[29] (hydrophilic–hydrophilic, hydrophilic–hydrophobic, hydrophobic–hydrophilic) immersed in pure water show that the distance between adjacent layers lies in the 0.24–0.36 nm range.

We conclude that the differences cannot be explained in terms of the interaction of water molecules with the 2D materials surface. These findings enable us to propose that on mildly-to-highly hydrophobic 2D materials surfaces immersed in water, the water molecules are expelled from the vicinity of the surface and replaced by two to three hydrophobic layers.

## Results

**Three-dimensional (3D)-AFM image of 2D materials–water interfaces.** Figure 1 shows some 3D-AFM images of the 2D material–water interface for graphene (Fig. 1a), few-layer $MoS_2$ (Fig. 1b) and few-layer $WSe_2$ (Fig. 1c). Images for $MoSe_2$ and $WS_2$–water interfaces are provided in the Supplementary Information (Supplementary Fig. 2). The 3D-AFM images show an alternation of darker and bright stripes that could reach 2 nm from the surface. To facilitate the interpretation of the data we include a 3D-AFM image obtained on a mica–water interface (Fig. 1d). Mica is an archetypical hydrophilic surface that has been extensively studied by 3D-AFM[30–34]. The comparison between the 3D-AFM images will also illustrate key differences between the organization of water on hydrophobic and hydrophilic surfaces. We have also measured the water contact angle on those interfaces. The values are in the 61 to 76° range (Supplementary Fig. 3). Those values indicate a moderately hydrophobic response. In contrast, the water contact angle measured on mica is zero, which underlines its hydrophilic character.

The quantitative features of the interfaces are readily visualized by plotting the 2D-AFM $xz$ maps (Fig. 2a–c). Those maps are extracted from the 3D-AFM images as a function of the $y$-value. The observables (phase shifts ($\phi$) and amplitudes ($A$)), have been transformed into force–distance curves (Fig. 2d–f) by using the force reconstruction methods[35–37] developed for amplitude modulation AFM (Supplementary Figs. 4 and 5). To increase the signal-to-noise ratio in the force–distance curves we have calculated the value of the force by averaging the values of the observables Obs for the different $x$ positions at the same $z$, this is,

$$\langle \text{Obs}(z) \rangle = \frac{\sum_1^n \text{Obs}(x_i, z)}{n} \qquad (1)$$

This approach enables us to detect changes in the tip-surface force of 20 pN.

To simplify the discussion, we apply the solvent-tip-approximation model to relate the features observed in the 2D force maps with changes in the solvent density[38]. From the force–distance curves we can deduce the position of the different solvation layers with respect to the position of the solid surface. We fix the position of the solid surface ($z = 0$) as the $z$-piezo displacement distance at which the repulsive force equals the maximum of the repulsive force associated with the closest solvation layer.

**Force–distance curves from 3D images.** Figure 2d–f show the force–distance curves obtained, respectively, for graphene, $MoS_2$ and $WSe_2$. The force–distance curves are characterized by the presence of several oscillations where the total force changes from

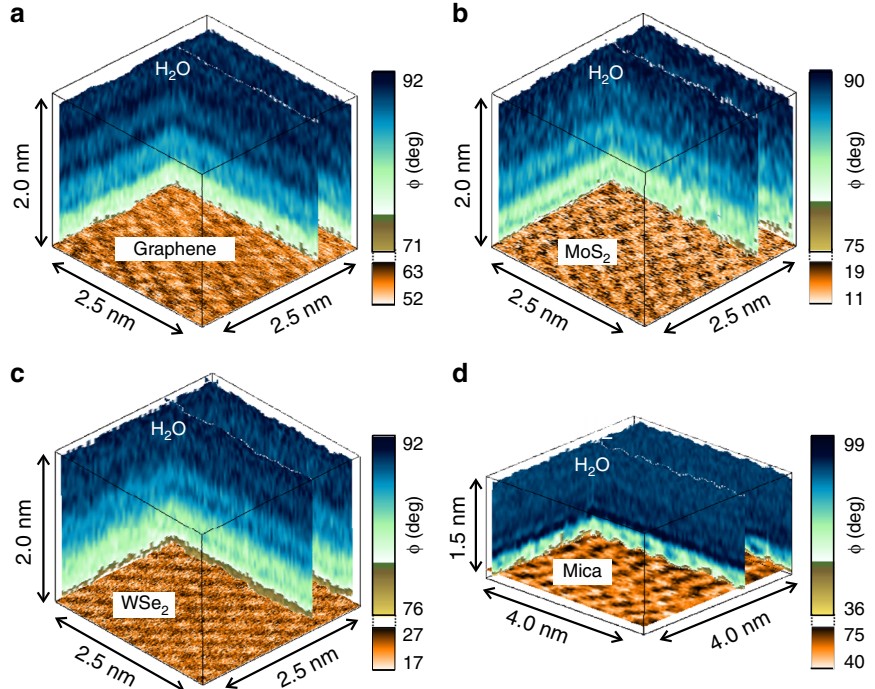

**Fig. 1** Three-dimensional (3D)-AFM images of 2D materials and mica–water interfaces. **a** Epitaxial graphene grown on SiC. **b** Few-layer $MoS_2$. **c** Few-layer $WSe_2$. **d** Mica. The images show the changes of the phase shift as function of the *xyz* position. There is an alternation of dark and light stripes. The oscillations are related to changes in the solvent density. Those changes give rise to a layered structure with a few-angstroms features. The interfacial structure is deeper on the 2D materials (hydrophobic surfaces) than on mica (hydrophilic). At the bottom of the 3D images we have plotted the lattice resolution images of the different surfaces (Supplementary Fig. 1)

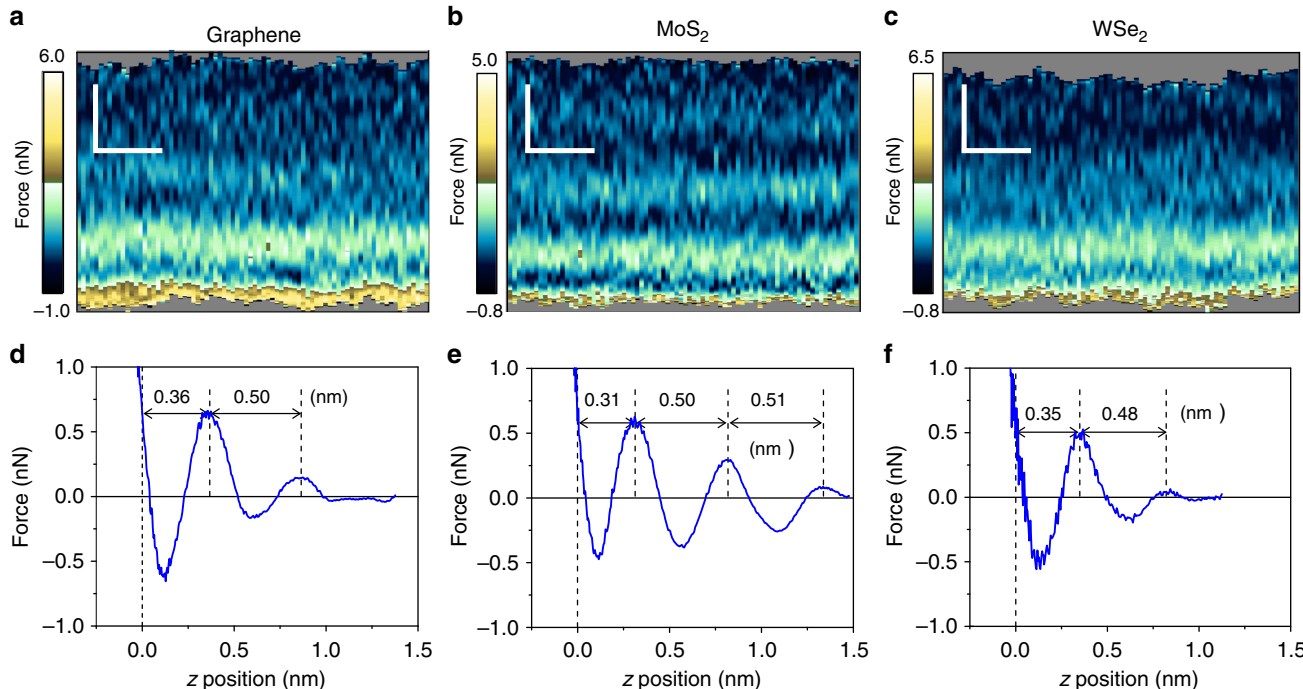

**Fig. 2** Two-dimensional (2D)-AFM *xz* force maps and force–distance curves of 2D materials–water interfaces. **a** Graphene. **b** Few-layer $MoS_2$. **c** Few-layer $WSe_2$. Scale bars represent 0.5 nm (both horizontally and vertically). **d** Force–distance curve for the graphene–water interface (extracted from **a**). **e** Force–distance curve for the $MoS_2$-water (extracted from **b**). **f** Force–distance curve for the $WSe_2$–water interface (extracted from **c**). The data show some oscillations in the force. Each peak is associated with a single molecular layer

positive (repulsive) to negative (attractive) values until the tip enters into mechanical contact with the surface. The number of the oscillations (2–3 peaks) obtained on the 2D materials and those on mica reveal that the extension of the interface is larger on the 2D materials (Fig. 3). This observation correlates with the hydrophobicity of the surfaces, moderately hydrophobic (2D materials) to highly hydrophilic (mica). The distance of the closest layer to the surface and the separation between adjacent layers have some slight dependence on the material. However, the most remarkable observation is that the interfacial distances measured on the 2D materials surfaces are about 0.2 nm larger than those measured on mica (Fig. 3b). Those differences cannot be explained in terms of the interaction of 2D materials with water molecules. The data extracted from the force–distance curve regarding the features of the interfacial water structure are summarized in Table 1.

We remark that the $z$ distance at which a maximum in the force is observed is shifted with respect to the distance of the corresponding maximum in the solvent density. The maxima of

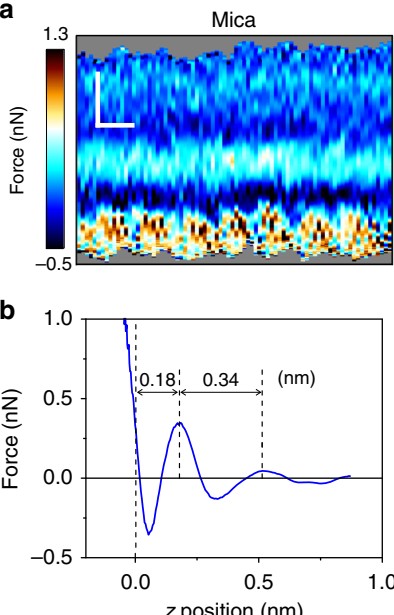

**Fig. 3** Two-dimensional (2D)-AFM $xz$ force map and force–distance curve of mica–water interface. **a** 2D-AFM map. Scale bars represent 0.5 nm (horizontally) and 0.2 nm (vertically). **b** Force–distance curve extracted from **a**). The mica is immersed in a 200 mM aqueous solution of KCl

**Table 1 Relevant distances extracted from the 2D-AFM $xz$ force maps**

|  | $d_0$ (nm) | $d_1$ (nm) | $d_2$ (nm) |
|---|---|---|---|
| Graphene | (0.36 ± 0.04) | (0.50 ± 0.04) | — |
| MoS$_2$ | (0.31 ± 0.02) | (0.50 ± 0.04) | (0.51 ± 0.03) |
| MoSe$_2$ | (0.30 ± 0.02) | (0.45 ± 0.05) | (0.47 ± 0.04) |
| WSe$_2$ | (0.35 ± 0.02) | (0.48 ± 0.03) | — |
| WS$_2$ | (0.36 ± 0.04) | (0.51 ± 0.05) | (0.62 ± 0.05) |
| Mica | (0.18 ± 0.01) | (0.34 ± 0.04) | — |
| Graphite | (0.44 ± 0.05) | (0.55 ± 0.03) | — |

$d_0$ is the distance to the solid surface; $d_1$ and $d_2$ are, respectively, the distances between the 1st and 2nd, and 2nd and 3rd adjacent hydrophobic layers Confidence intervals represent the standard deviation

the forces happen at larger distances. The solvent-tip approximation explains this observation by establishing that the force is proportional to the logarithmic derivative of the solvent density[8,38,39].

For graphene-water we observe a first layer at height of 0.36 nm from the epitaxial graphene plane, a second layer appears at 0.86 nm from the surface. We have also performed experiments on graphite–water interfaces (Supplementary Fig. 6). On graphite we observe two layers located, respectively, at heights of 0.44 and 0.99 nm from the graphite surface. Those values are close to the distances measured on graphene (the difference is of tens of pm). An early AFM experiment reported that the distances between layers on graphite were larger than on mica[40].

For the few-layer MoS$_2$ and WSe$_2$–water interfaces the first layer appears at a height of, respectively, 0.31 and 0.36 nm from the surface. The 2nd layers are observed at, respectively, 0.81 and 0.83 nm. In some cases (Fig. 2b) we observe a 3rd layer at a height of 1.31 nm. The distances between the first two adjacent layers are for MoS$_2$ and WSe$_2$, respectively, 0.50 and 0.48 nm. Those values are significantly larger than the van der Waals diameter of a water molecule (0.28 nm). MD simulations performed for a wide range of crystalline materials such as graphene[15], MoS$_2$[41], p-nitroaniline[42], calcite[43] or α-Al$_2$O$_3$[44] immersed in pure water show that the distance between adjacent hydration layers hardly depends on the nature of the surface. The values range from 0.28 to 0.33 nm. On the other hand, the distances between adjacent layers measured on graphene (0.50 nm), MoS$_2$ (0.50 nm) and WSe$_2$ (0.48 nm) are comparable to the values reported for some crystalline surfaces (p-nitroaniline[42] (0.44 nm) and graphite[45] (0.60 nm)) immersed in hydrocarbon solvents. In those cases, MD simulations show a correlation between the distances measured between adjacent layers and the molecular-size of the solvent[42]. Based on the above data, we conclude that the interfacial structure observed on 2D materials–water interfaces cannot be solely explained in terms of the interaction of water molecules with the 2D materials. This leads the search for another chemical species.

## Discussion

The experiments have been performed with purified water (see methods). However, once a purified water–air interface is established, gas molecules and/or trace airborne contaminants are dissolved in the liquid water. In fact, the adsorption of airborne molecules on graphitic surfaces that have been exposed to air and/or water has been the subject of several studies. It is known that airborne hydrocarbon contaminants are deposited on hydrophobic surfaces. Those contaminants modify several properties of graphitic surfaces[18,46,47], in particular, they increase the water contact angle value[18]. The wettability of graphene influences other properties such as adhesion or carrier mobility. Hydrocarbon contaminants are composed of alkanes, alkenes, alcohol, and aromatic species. Those contaminants are ubiquitous in ambient air (few parts per trillion)[47]. Airborne contaminants are spontaneously incorporated into purified water from ambient air. Those contaminants could be segregated from the water to form the hydrophobic layers.

Gas molecules (N$_2$, O$_2$) could also be adsorbed in the liquid water from the ambient air[48,49]. Hwang and co-workers[50] observed the growth of molecular structures on graphite surfaces immersed in water. They proposed that the observed structures were composed of condensed gas (N$_2$) molecules adsorbed from the air[50–52]. High-resolution AFM images of different epitaxial graphene interfaces in air showed the presence of ordered and disordered adsorbates on the graphene surface[53,54]. We have also observed those structures (Supplementary Figs. 7 and 8).

Wastl et al.[54] pointed out that the ordered (stripe) structure observed in air was similar to the stripe structure observed on graphite samples immersed in water[51]. More recently, Schlesinger and Sivan[55,56] have proposed that the solvation structure observed on graphite by 3D-AFM was dominated by the layering of condensed gas molecules. They have also claimed that the above layer structure is a general property of hydrophobic surfaces in contact with water. It has also been reported that graphite immersed in water could spontaneously catalyze methanol from carbon dioxide present in the liquid water[57,58]. However, this chemical reaction has not been described for few-layer transition metal dichalcogenides, which makes this process an unlikely source of the chemical species present in the hydrophobic layers.

In short, the experimental evidence supports the formation of molecular-thick hydrophobic layers composed from molecules present in the ambient air and dissolved into the liquid water. The chemical composition of the adsorbates on graphene and graphite–air interfaces has been characterized by different spectroscopy methods[46,47]. Those methods confirm the presence of airborne hydrocarbon contaminants. Unfortunately, those measurements are hard to perform on solid surfaces covered by liquid water. Our data slightly favors airborne hydrocarbon molecules as the main component of the hydrophobic layers. First, we observe a correlation between the aging of the surface (time of exposure to air before immersion in water) with the increase of the water contact angle (Fig. S2). Second, the easiness to observe the layered structure also correlates with the aging of the surface. Third, the distances measured between hydrophobic layers are similar to the ones measured on graphite and on organic crystals (p-nitroaniline)[42] immersed in organic solvents (0.48–0.50 nm versus 0.44 nm). Four, some theoretical calculations show that the adsorption energy for small adsorbates such as $N_2$ is smaller than the adsorption energy for larger airborne hydrocarbon contaminants[20]. Those calculations predict that at room temperature the thicknesses for hydrocarbon and $N_2$ layers are, respectively, 0.98 and 0.27 nm.

Notwithstanding the above arguments, the chemical nature of the hydrophobic layers observed in liquid water, either hydrocarbon contaminants or condensed gas molecules, can only be settled by performing high-resolution spectroscopy measurements.

Lastly we compare the interfacial structures for mica and 2D materials. On mica (Fig. 3) the layered structure is in good agreement with the distances predicted by MD simulations for hydrophilic surfaces immersed in pure water (~0.30 nm). On 2D materials–water interfaces we observed a layered structured characterized by separations between adjacent layers of 0.5 nm range. Those values cannot be explained in terms of properties of pure water. The paradox is resolved by analyzing the hydrophilic–hydrophobic interactions between the solid surface, the water and hydrophobic contaminants. On mildly hydrophobic surfaces (graphene, few-layer $MoS_2$, and $WSe_2$) the contaminants dissolved in the water diffuse to the hydrophobic interface where they displace the water molecules to reduce the free energy. On mica this process does not reduce the free energy because the water molecules are attracted to the negatively charged mica surface. There, the hydrophobic contaminants are expelled from the interface.

We remark that the experiments have been performed with purified water. The incorporation of gas molecules and/or trace airborne contaminants into the water is an unavoidable process once a water–ambient air interface is formed.

In Fig. 4 we summarize the observations and our interpretation. The schemes highlight the differences of the interfacial water structure of graphene, few-layer $MoS_2$, few-layer $WSe_2$ and mica. The interaction of the liquid water molecules with a 2D material

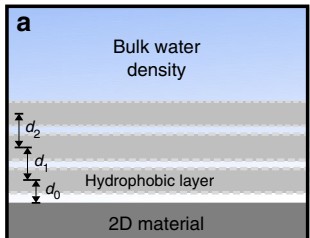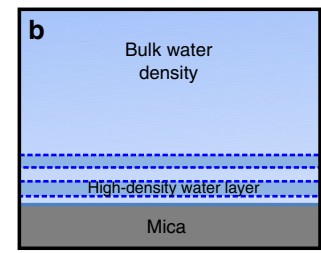

**Fig. 4** Schemes of the interfacial layering on 2D materials and mica surfaces. **a** 2D materials. The water molecules are expelled from the vicinity of the surface. They are replaced by several hydrophobic layers. These layers are composed of airborne molecular species. The spatial period is about 1.7 times larger than the diameter of a water molecule (~0.30 nm); $d_0$ is the distance to the solid surface; $d_1$ and $d_2$ are, respectively, the distances between the 1st and 2nd, and 2nd and 3rd adjacent hydrophobic layers. **b** Mica. The water density oscillates around the value of the bulk water density. The periodicity is close to the diameter of a water molecule

surface gives rise to a pronounced structuring of the interface characterized by the presence of 2 to 3 hydrophobic layers.

## Conclusion

Three-dimensional AFM reveals that the interfacial structure of a 2D materials–water interface is characterized by a layered structure formed by hydrophobic layers. The hydrophobic layers are composed of molecules coming from the air and dissolved into the liquid water. The hydrophobic molecules minimize the free energy of the interface by displacing the water molecules from the 2D materials surface. These results underline that the hydrophobicity of 2D materials has singular features at the atomic and molecular level. The existence of hydrophobic layers on the vicinity of 2D materials surfaces should influence the interactions of those materials with molecules, salts or proteins present in the water.

3D-AFM has provided experimental images of the existence of molecular-size hydrophobic layers on graphene, $MoS_2$ (few and thick layer) and $WSe_2$ (few and thick layer) and graphite. Those surfaces have different chemical, electronic, and optical properties but they share some common features such as the existence of atomically flat terraces and their hydrophobic character. Altogether, these results underline that the formation of molecular-size hydrophobic layers is a universal property that applies to any atomically flat hydrophobic surface immersed in liquid water equilibrated with ambient air.

## Methods

**Two-dimensional (2D) materials, graphite, and mica**. Thin flakes of $MoS_2$ (SPI supplies, USA), $WSe_2$, $MoSe_2$, and $WS_2$ (all from HQ Graphene, Netherlands) were mechanically exfoliated with adhesive tape and transferred onto the clean Si/SiO2 substrates. These few-layer flakes have thicknesses in the 5–50 monolayers range. Epitaxial graphene monolayers (98 ± 5% monolayer coverage) on the silicon face of SiC substrates were purchased from Graphene Nanotech SL, Spain. The graphene is grown through selective sublimation of Si surface atoms by high-temperature annealing. The sample was cleaned by rinsing it with ultrapure water before the experiments. HOPG (grade ZYB) samples were purchased from Bruker (USA) and cleaved with adhesive tape before the experiment. Discs of muscovite mica (Grade V-1, SPI supplies, USA) were freshly cleaved with adhesive tape before the experiments and copiously rinsed with ultrapure water.

**Water**. Ultrapure water was freshly obtained before the experiments (ELGA Maxima, 18.2 MΩ). A few minutes after its purification, the water's pH was measured. It reached a value of 5.6 (Hanna Instruments HI 9024). The solutions of 200 mM KCl were prepared with KCl salt (≥99.0%, Sigma-Aldrich) dissolved in the ultrapure water.

We have performed some 3D-AFM experiments on a few-layer $MoS_2$ surface immersed in water mixed with *n*-octane (≥99%, Sigma-Aldrich) at a concentration of $c_{n\text{-octane}} = 0.7$ mL: 1 L. The above concentration corresponds to the solubility limit of octane in water (Supplementary Fig. 9).

**Cleaning protocols**. The few-layer transition metal dichalcogenides were deposited on Si/SiO$_2$ (275 nm SiO$_2$, thermally oxidized) substrates. The substrates were sequentially ultrasonicated in acetone (99.6%, Acros Organics), ethanol (≥99.8%, Sigma-Aldrich), and ultrapure water. After drying the substrates with a flow of nitrogen (gas), they were then exposed to oxygen plasma for 15 min (Diener Electronic, Germany).

In the experiments performed with thick samples (HOPG, WSe$_2$, MoS$_2$), the sample was cleaved while being immersed in water to avoid any contact with the ambient air before 3D-AFM imaging. To this end, we glued a small piece of each material on a Teflon disc and pressed adhesive tape against the dry surface. The sample was immersed into fresh ultrapure water contained in a wide glass beaker and we pulled off the tape while being immersed.

**AFM sample stage**. After the cleaning protocol, the samples were mounted onto the microscope sample stage. Then, a small droplet of ultrapure water or an electrolyte solution (in the case of mica) was placed on the surface and the cantilever was immersed into the liquid. The experiments were performed in a closed liquid cell, where the temperature was held constant at $(28.0 \pm 0.1)$ °C.

**AFM imaging**. Conventional AFM images of the 2D materials, mica, and bulk samples were performed before 3D-AFM imaging to localize clean and atomically flat areas of the samples (Supplementary Fig. 8a, b). Those AFM images were obtained in the amplitude modulation (AM) mode by exciting the second mode of the cantilever. The free amplitudes $A_0$ were typically of 240 pm, with an amplitude set-point $A_{sp} = 0.90\ A_0$. Examples of such scans are shown in Supplementary Figs. 1 and 2. Those measurements were done in a commercial instrument (Cypher VRS, Asylum Research, Oxford Instruments).

**Three-dimensional (3D)-AFM**. The three-dimensional AFM concept[27] was implemented on a Cypher VRS platform (Asylum Research, Oxford Instruments). Original codes were developed to control the motion of the tip and the feedback electronic circuits. Three-dimensional AFM imaging is performed in the amplitude modulation mode by exciting the microcantilever at its 2nd eigenmode. At the same time that the cantilever oscillates with respect to its equilibrium position, a sinusoidal signal is applied to the $z$-piezo to modify the relative $z$ distance between the sample and the tip. We have used $z$-piezo displacements with amplitudes between a 1 and 4 nm and a period (frequency) of 20 ms (50 Hz). (Supplementary Figs. 4 and 10). The $z$-piezo signal is synchronized with the $xy$ displacements in such a way that for each position on the surface of the material, the tip performs a single and complete $z$ displacement.

The oscillation of the cantilever was driven by photothermal excitation. The free amplitude values $A_0$ were in the 40–360 pm range. Supplementary Fig. 4 shows the dependence of the amplitude on the tip-sample distance for the different solid–water interfaces. The feedback monitors the instantaneous amplitude and acts on the $z$-piezo to keep the amplitude at a fixed value ($A_{sp} \approx 0.90\ A_0$). We have used a relatively small feedback bandwidth (1 kHz). This means a feedback response that is too slow to respond to the applied $z$-piezo displacement signal but fast enough to track the topography of the surface.

The $z$ data is read out every 20 μs and stored in 1024 pixels (512 pixels half cycle). Each $xy$ plane of the 3D map contains $80 \times 64$ pixels. Hence, the total time to acquire such a 3D-AFM image is 105 s.

**Microcantilevers**. Silicon cantilevers with silicon tips were used for 3D-AFM imaging (PPP-NCHAuD, NanoAndMore, Germany). The cantilevers were cleaned first in a mixture (50:50 in volume) of isopropanol (99.6%, Acros Organics) and ultrapure water, rinsed with ultrapure water and then placed in a UV-Ozone cleaner (UV3, Novascan Technologies, USA) for ≈1 h to remove organic contaminants.

To prevent tip damage during the calibration, the cantilevers were calibrated in liquid once the 3D-AFM data was acquired. The calibration of the force constant includes the following protocol. First, the inverted optical lever sensitivity (invOLS) for the static deflection $\sigma_0$ is determined. This step involves the measurement of force–distance curves on a clean, flat, and stiff sample (275 nm of thermally grown SiO$_2$ on Si). The invOLS is determined from the curve's slope in the contact part. Second, a cantilever's thermal noise spectrum (PSD) is recorded at about 2 μm above the sample surface. We fit the single harmonic oscillator (SHO) model to the PSD around the peak of the first resonance frequency using the calculated invOLS of the first eigenmode, $\sigma_1 = 1.09\sigma_0$[59]. From the fitting we obtain the force constant $k_1$, quality factor $Q_1$, and resonance frequency $f_{r1}$. Then we measure the resonance frequency of the second eigenmode, $f_{r2}$, from the PSD and calculate the corresponding force constant $k_2 = k_1(f_{r2}\ f_{r1}{}^{-1})^{2.17}$, as proposed by Labuda et al.[60]. Knowing $k_2$, we then fit the SHO model to the PSD around the peak of the second resonance frequency to obtain the corresponding invOLS $\sigma_2$ and $Q_2$. The force constants used to determine the forces are summarized in Supplementary Table 1.

Additional 3D-AFM experiments have been performed with other cantilevers (different types). No major differences have been observed.

## Data availability

All supplementary figures and tables referenced in the main text are provided in the supplementary information. The raw data used to create the figures is freely available from the open-source data repository Figshare. The files can be accessed through https://doi.org/10.6084/m9.figshare.8157899.

## Code availability

The code used to analyze the data is available on request.

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

## Acknowledgements

This work was funded by the European Research Council ERC-AdG-340177 (3DNa-noMech) and the Ministerio de Economía, Industria y Competitividad (MAT2016-76507-R).

## Author contributions

M.R.U. performed the experiments on the 2D materials and processed all the data. D.M.-J. performed the experiments on mica. R.G. conceived the project, planned the experiments, and wrote the MS. M.R.U. and R.G. discussed the data. M.R.U., D.M.-J. and R.G. read and revised the MS.

## Additional information

**Competing interests:** The authors declare no competing interests.

