## [Peer Review File · Nature Communications]

Reviewers' comments:

Reviewer #1 (Remarks to the Author):

Uhlig and coworkers report three-dimensional (3D) atomic force microscopy (AFM) measurements of moderately hydrophobic materials (graphene, MoS₂, WSe₂, and graphite) immersed in ultra-pure water.

The authors prepared specimens very carefully, and find that the separation of local maxima in force curves recorded over the hydrophobic surfaces were about 0.50 nm, which is clearly larger than that recorded over a hydrophilic surface (mica; about 0.30 nm).

Based on comparisons with previous literature of hydrophobic and hydrophilic surfaces, they concluded that the characteristic oscillating force response of the immersed hydrophobic surfaces is attributed to "hydrophobic layers" composed of inevitable hydrocarbon contaminants (or maybe dissolved N₂ molecules) in water.

Molecular arrangements of interfaces between hydrophobic materials and water is a long-standing unsolved problem and this topic may attract wide-ranging interests. However, the results and conclusions in this manuscript (MS) are basically same as a previous report by Yang et al. (PCCP 20 (2018) 23552; ref. 23 in MS). Yang et al. simultaneously measured 3D-AFM of hydrophobic (graphene) and hydrophilic (mica) surfaces immersed in water and found the difference in the force curves. They already referred to the effect of N₂ and hydrocarbon contaminants on the force response at the hydrophobic material.

In MS, therefore, the analysis and identification of the hydrophobic layers are insufficient. To obtain new evidence of the hydrophobic layers, further investigations (for example, time dependence of AFM images, force curves, and water contact angles for each material, 3D-AFM recorded over the ordered structures with a stripe pattern (shown in Fig. S1-S2) as well as Yang et al., and force measurements of materials immersed in highly contaminated water) are required. For the above reason I cannot recommend publication of this manuscript in Nature Communications, and it is better suited for a specialized journal.

Reviewer #2 (Remarks to the Author):

In this manuscript, the 3D AFM was employed to map the hydrophobic layers on graphene, few-layer MoS₂ and WSe₂ in water at angstrom resolution, which provides a novel way to understand the interface wetting behavior of water. This work is quite interesting and very exciting results. It will generate a lot of interests from many disciplines and benefits the readership of Nature Com..

Although the authors obtained some original results, other experimental conditions should also be explored. I would suggest the acceptance of this manuscript after addressing the following points.

1. The interfacial water organized on graphene, few layer MoS₂ and few layer WSe₂ was investigated by 3D AFM, will the thickness (for example, monolayer or bulk of MoS₂ and WSe₂) of the 2D nanomaterials affect the distance between adjacent layers and the thickness of the hydrophobic layers?

2. Like the liquid, solid surface also has surface tension and surface energy. In order to minimize the free energy of the interface, some molecules in air or liquid can adsorb on the solid surface spontaneously. It's the phenomenon that the authors reported. Is this adsorption a physical adsorption? Will the temperature affect the hydrophobic layers?

3. On page 11, the authors concluded that 'The hydrophobic layers are composed of molecules coming from the air and dissolved into the liquid water'. Will the environment changes, especially the air environment, have some impacts on the experimental results? How to control the experimental conditions and assure the repeatability of the experiment?

4. Whether the composition of the solution (not pure water) can affect the structure of the hydrophobic layers?

5. For the high-resolution reconstruction in this manuscript, whether the lateral and vertical

thermal drift in liquid has impacts on the precise of the data?

6. The authors should check the manuscript more carefully, there are some small errors. For example, the figure label e, f, g marked in Figure 2 should be d, e, f.

Reviewer #3 (Remarks to the Author):

In this paper, authors used a 3D-AFM to characterize the three-dimensional structure of water near the surface of graphene and few-layer MoS₂ and WSe₂. They reported an oscillating structure characterized by the presence of up to three solvation layers within the last 2 nm of the liquid. On hydrophobic 2D materials surfaces immersed in water, the water molecules are expelled from the vicinity of the surface and replaced by two to three hydrophobic layers. The procedure provided in this paper is very helpful for us to understand the structure of water near the 2D material surface. This work is very interesting and is written very well.

Why the distances between the first two adjacent layers for MoS₂ and WSe₂ are larger than the van der Waals diameter of a water molecule?

Does the rising of temperature change the behavior of water molecules and the number of oscillations? Authors have measured the water contact angle on interfaces. The contact angle will change with temperature.

Authors should provide an uncertainty analysis of your observation.

Reviewers questions are highlighted in bold.

Reviewer: 1

Uhlig and coworkers report three-dimensional (3D) atomic force microscopy (AFM) measurements of moderately hydrophobic materials (graphene, MoS₂, WSe₂, and graphite) immersed in ultra-pure water. The authors prepared specimens very carefully, and find that the separation of local maxima in force curves recorded over the hydrophobic surfaces were about 0.50 nm, which is clearly larger than that recorded over a hydrophilic surface (mica; about 0.30 nm). Based on comparisons with previous literature of hydrophobic and hydrophilic surfaces, they concluded that the characteristic oscillating force response of the immersed hydrophobic surfaces is attributed to "hydrophobic layers" composed of inevitable hydrocarbon contaminants (or maybe dissolved N₂ molecules) in water.

1. Molecular arrangements of interfaces between hydrophobic materials and water is a long-standing unsolved problem and this topic may attract wide-ranging interests. However, the results and conclusions in this manuscript (MS) are basically same as a previous report by Yang et al. (PCCP 20 (2018) 23552; ref. 23 in MS). Yang et al. simultaneously measured 3D-AFM of hydrophobic (graphene) and hydrophilic (mica) surfaces immersed in water and found the difference in the force curves. They already referred to the effect of N₂ and hydrocarbon contaminants on the force response at the hydrophobic material. In MS, therefore, the analysis and identification of the hydrophobic layers are insufficient.

Reply. We are perplexed by the comment regarding the novelty of the data. We disagree. During the performance of the experiments and before writing the Ms. we did a thorough search of the scientific literature. We read and cited several contributions from Hwang's group, in particular, four of his papers have been cited (refs 23 PCCP 20, 23522 (2018) and refs. 50-52).

In the following we explain/show/demonstrate that the materials, topic and scope of this Ms. are different from Yang et al. paper.

Materials. Figure 3b in Yang et al. (PCCP 20 (2018) 23522; ref. 23 in MS) shows a step height of about 5 nm. Bianco et al. (Carbon 65, 1-6 (2013)) provide the definitions of graphene (monolayer), few-layer graphene (2 to 5), and multi-layer graphene (below 10 monolayers). Those definitions are currently used by the 2D materials community. The graphitic sample used in Yang al. experiments indicates that the number of layers is about 15. This sample is indeed a very thin **graphite (HOPG)** flake. This sample will show the same intrinsic mechanical, optical, chemical or electronic properties of a bulk HOPG sample.

Yang et al. studied a 5 nm thick graphite (HOPG) flake. We have studied more than 10 different hydrophobic surfaces. We have devoted a significant amount of time and effort to fabricate and select few-layer and multi-layer flakes.

Topic & scope. Copied from the abstract of Yang et al. *'Here, we prepared hydrophilic mica substrates with some areas covered by mildly hydrophobic graphene layers and studied the resulting*

hydration layers using three-dimensional (3D) force measurements based on frequency-modulation atomic force microscopy.'

First, the sample in Yang's et al experiments is neither graphene nor a graphene multilayer (see above). In addition, Yang *et al.* do not provide a single argument that could support the extension of the observations from a very thin HOPG sample to 2D and few-layer materials interfaces. After all, the emergence of 2D and few-layer materials is based on the variety of the mechanical, chemical, electronic and optical properties that those materials have (refs. 10-13).

Novelty of the data. We provide **the first atomic resolution 3D images** of graphene (monolayer) and a few-layer MoS₂ and WSe₂ immersed in water. Since the first submission, we have also obtained atomic resolution 3D images of a few-layer MoSe₂ and WS₂. The new data (Supplementary information) strengthen the conclusions of the manuscript.

Novelty of the conclusions. Our data **strongly supports** that the formation of molecular-size hydrophobic layers is a **universal** property that applies to any atomically flat hydrophobic surface (of the 2D materials family) immersed in liquid water equilibrated with ambient air. Our data indicates that the same behaviour should be observed on any flat hydrophobic surface. The conclusions are supported by performing a comprehensive series of 3D-AFM experiments. We have performed more than 100 experiments on 11 different surfaces: MoS₂ (few-layer, multilayer, bulk), WSe₂ (few-layer, multilayer, bulk) MoSe₂ (few-layer, multilayer, bulk), WS₂ (few-layer, multilayer, bulk), graphene, graphite and mica. The **reproducibility** of the observations enables us to formulate the universality of the implications.

2. To obtain new evidence of the hydrophobic layers, further investigations (for example, time dependence of AFM images, force curves, and water contact angles for each material, 3D-AFM recorded over the ordered structures with a stripe pattern (shown in Fig. S1-S2) as well as Yang et al., and force measurements of materials immersed in highly contaminated water) are required. For the above reason I cannot recommend publication of this manuscript in Nature Communications, and it is better suited for a specialized journal.

Reply. We have performed the 3D-AFM experiments suggested by the Reviewer. Those experiments provide additional support to the findings reported in the manuscript. Specifically we discuss below the experiments on the graphene ripples and by using contaminated water.

-3D-AFM on the stripe patterns.

Figure R1a shows an AFM phase image of a stripe pattern region of the graphene surface (ripple). Figure R1b shows a 2D-AFM xz map obtained from a 3D AFM measurement on the region showed in (a). The structure of the solid-liquid interface observed on the ripple region is identical to the one observed on the flat graphene regions

Figure R1. a. AFM phase image of ripples on a graphene surface immersed in purified water. b. 3D-AFM image obtained on the very same region shown in a. c. xy image extracted from the 3D-AFM image shown in b ($z = 0.05$ nm, i.e., underneath the hydrophobic layers). The image shows the ripple structure. d. 2D-AFM xz force map of the graphene ripple-water interface extracted from a 3D-AFM measurement.

-Experiments using highly contaminated water

We have performed some 3D-AFM experiments on a few-layer MoS₂ surface immersed in water mixed with *n*-octane ($c_{n\text{-octane}} = 0.7 \text{ mL} : 1 \text{ L}$). Figure R2 shows a 2D-AFM xz force map. The above concentration corresponds to the solubility limit of octane in water. It is the highest possible concentration. This liquid fits the definition of highly contaminated water.

Figure R2. a 2D-AFM xz force map of the MoS₂-octane:water interface extracted from a 3D AFM measurement.

The image shows the characteristic alternation of dark and light stripes observed in purified water. We observe up to 4 hydrophobic layers. The distances between the first two adjacent layers coincide with those obtained with purified water (see Table 1). The 3D-AFM data indicate that the number of hydrophobic layers increases when a few-layer MoS₂ surface is immersed in water that has been mixed with a liquid hydrocarbon (*n*-octane). This result strengthens a key finding of the manuscript: in the vicinity of a hydrophobic surface, water is expelled from the interface and replaced by hydrophobic layers.

Reviewer: 2

In this manuscript, the 3D AFM was employed to map the hydrophobic layers on graphene, few-layer MoS₂ and WSe₂ in water at angstrom resolution, which provides a novel way to understand the interface wetting behavior of water. This work is quite interesting and very exciting results. It will generate a lot of interests from many disciplines and benefits the readership of Nature Com.. Although the authors obtained some original results, other experimental conditions should also be explored. I would suggest the acceptance of this manuscript after addressing the following points.

1. The interfacial water organized on graphene, few layer MoS₂ and few layer WSe₂ was investigated by 3D AFM, will the thickness (for example, monolayer or bulk of MoS₂ and WSe₂) of the 2D nanomaterials affect the distance between adjacent layers and the thickness of the hydrophobic layers?

Reply. The 3D-AFM experiments performed on graphene (single layer) and graphite flakes (hundreds of layers) and those on MoS₂ (few-layer, multi-layer and bulk) do not show any significant differences in the distances between hydrophobic layers. We observe some minor changes on the distance between the first hydrophobic layer and the solid surface. Smaller distances are observed on graphene (0.36 nm) than on the bulk material (0.44 nm).

2a. Like the liquid, solid surface also has surface tension and surface energy. In order to minimize the free energy of the interface, some molecules in air or liquid can adsorb on the solid surface spontaneously. It's the phenomenon that the authors reported. Is this adsorption a physical adsorption?

Reply. Indeed, this is in part the process we are referring to. We consider that the molecules are physisorbed to the surfaces. Adsorption of organic molecules from non-polar solvents onto graphitic surfaces has a long history. In particular, n-alkanes and alkane derivatives such as alcohols form well-ordered structures on HOPG surfaces. This has been observed by STM (e.g. Rabe et al. Science 1991, 253, 424) and AFM (e.g., Hiasa et al. J. Phys. Chem. C 2012, 116, 26475 and ref. therein). In these cases, the adsorbate-substrate interaction is of physical origin, governed by van der Waals forces between the alkyl chains and the surface carbon. A good introduction to the free energy considerations we are implying can be found in J. N. Israelachvili, Intermolecular and Surface forces, 3rd edition, Chapter 17.

2b. Will the temperature affect the hydrophobic layers?

Reply. The temperature should affect the layering because the energy barrier is weighted by a Boltzmann factor ($\exp(-U/k_B T)$), where U is the activation barrier for the desorption of a hydrophobic molecule. By increasing the temperature, the hydrophobic molecules should be removed from the surface. Our experimental set-up only allows changing the temperature between 28 and 33 °C. In this range, we have not observed significant changes on the structure of the hydrophobic layers (Fig. R3)

Figure R3. 2D AFM xz force maps of the graphene-water interface at different temperatures. a. 2D AFM xz force map at 28 °C. b. 2D AFM xz force map at 32.5 °C

3a. On page 11, the authors concluded that ‘The hydrophobic layers are composed of molecules coming from the air and dissolved into the liquid water’. Will the environment changes, especially the air environment, have some impacts on the experimental results?

Reply. Water is equilibrated with ambient air. Even in the absence airborne hydrocarbon contaminants, air molecules (N_2 , O_2 and CO_2) will spontaneously diffuse into liquid water. The hydrophobic layers could be made of condensed N_2 molecules. See also Figure R2 and the associated text. Regarding this point we copy the paragraph from ref. 47 (Li *et al.* ACS Nano 10, 349-359 (2016): “It is extremely difficult to maintain a hydrocarbon-free environment because even a parts per trillion level of hydrocarbon is detrimental. Although it is possible to remove

hydrocarbons from air by passing contaminated air through cryogenically cooled activated charcoal,²⁷ such an approach requires the experimental setup to be isolated from ambient air, making it impractical for most experiments and large-scale applications. We also note that a glovebox and clean room do not provide a hydrocarbon-free environment; in fact, both contain high levels of hydrocarbon due to emission from plastics (e.g., gloves, wafer storage containers, etc.)”

3b. How to control the experimental conditions and assure the repeatability of the experiment?

Reply. The data and conclusions are very reproducible. They have been verified in more than 100 different experimental runs. The experiments were performed on 10 different hydrophobic surfaces such as MoS_2 , WSe_2 , $MoSe_2$, WS_2 and graphene (as well as in their bulk counter parts). Those surfaces were prepared by different methods, such as mechanical cleavage and

epitaxial growth (graphene on SiC). We have used water from three different purification systems (two identical ELGA Maxima systems and a home-made purifier).

4. Whether the composition of the solution (not pure water) can affect the structure of the hydrophobic layers?

Reply. We have performed some 3D-AFM experiments on a WSe₂ surface immersed in PBS (phosphate buffered saline, 10 mM) and KCl (200 mM). Figure R4 shows 2D-AFM *xz* force maps obtained in those aqueous solutions. The alternation of regions of high and low force values is very similar to the one obtained in purified water. However, the separation between the 1st and the 2nd adjacent layer seems larger in the present of salts (0.53 nm (10 mM PBS), 0.58 nm (200 mM KCl) vs. 0.48 nm (pure water)). These results are preliminary. They are based on a few measurements, however, they seem to indicate that the present of dissolved salts favours the separation of the hydrophobic layers. We acknowledge that this is a relevant scientific problem. However, it departs from the central points of the manuscript. We plan to investigate it in the near future.

Figure R4. a. 2D-AFM *xz* force map of WSe₂ immersed in a 10 mM PBS solution. b. 2D-AFM *xz* force map of WSe₂ immersed in a 200 mM KCl solution.

5. For the high-resolution reconstruction in this manuscript, whether the lateral and vertical thermal drift in liquid has impacts on the precise of the data?

Reply. The experiments have been performed at a constant temperature, usually, 28.0° C. The temperature of the AFM chamber can be fixed in the 28-33 °C range by a module that generates steps of ± 0.1 K. The thermal stability of the *xyz* scanner is of ± 0.015 K and its thermal drift is of 20 nm/K (Asylum Research, Oxford Instruments), that means, that once the working temperature has been reached, the thermal drift is of 0.3 nm (± 0.015 K \cdot 20 nm/K).

Figure R5 shows the thermal stability of the *xyz* position. We have monitored the *xyz* position over 12 hours. The tip has remained in the same position within a 0.3 nm incertitude.

Figure R5. Temperature and corresponding drift at the AFM scanner (Head) recorded over a time of 12 hours after equilibrating the system. The temperature set-point was set to 28.0 °C. The left axis shows the changes in the position caused by the thermal drift. The right axis shows the temperature of the chamber.

Each individual force-distance curves of a 2D-AFM xz force maps is acquired 10 ms. On this time scale, the temperature is constant. As a consequence, the thermal drift has no influence on our data.

The thermal drift could affect the lateral distances measured on the slow y axis. However, the effect is also negligible. For example, the time needed to acquire two adjacent xz frames separated in the y axis is 1.6 s.

6. The authors should check the manuscript more carefully, there are some small errors. For example, the figure label e, f, g marked in Figure 2 should be d, e, f.

Reply. We thank the referee for this observation. The errors have been corrected.

Reviewer: 3.

In this paper, authors used a 3D-AFM to characterize the three-dimensional structure of water near the surface of graphene and few-layer MoS₂ and WSe₂. They reported an oscillating structure characterized by the presence of up to three solvation layers within the last 2 nm of the liquid. On hydrophobic 2D materials surfaces immersed in water, the water molecules are expelled from the vicinity of the surface and replaced by two to three hydrophobic layers. The procedure provided in this paper is very helpful for us to understand the structure of water near the 2D material surface. This work is very interesting and is written very well.

1 Why the distances between the first two adjacent layers for MoS₂ and WSe₂ are larger than the van der Waals diameter of a water molecule?

Reply. The distance between the adjacent hydrophobic layers is larger than the van der Waals diameter of a water molecule because the interface is mostly composed of hydrophobic molecules. The distance between the first hydrophobic layer and the solid surface ranges between 0.30 nm and 0.36 nm. Those values are not far from the van der Waals diameter of a water molecule; however, this is a coincidence. Our data shows that water has been displaced from the interface. This distance is defined from the first maximum of the force curve to the sample surface. Hence this specific distance is *not* equivalent to the diameter of the involved molecules. It rather corresponds to the species' molecular radius plus an offset (depletion) that is surface-dependent. Figure R6 (Fig. 4a in the main text) provides a scheme of the definition of distances.

Figure R6. Scheme of the hydrophobic layers on 2D materials. d_0 is the distance to the solid surface; d_1 and d_2 are, respectively, the distances between the 1st and 2nd, and 2nd and 3rd adjacent hydrophobic layers.

2 Does the rising of temperature change the behavior of water molecules and the number of oscillations? Authors have measured the water contact angle on interfaces. The contact angle will change with temperature.

Reply. See also reply 2 to Reviewer 2. The temperature should affect the layering because the energy barrier is weighted by a Boltzmann factor ($\exp(-U/k_B T)$), where U is the activation barrier for desorption. By increasing the temperature, the hydrophobic molecules should be removed from the surface. Our experimental set-up only allows controlling the temperature between 28 and 33 °C. In this range we have not observed significant changes on the structure of the hydrophobic layers.

3 Authors should provide an uncertainty analysis of your observation.

Reply. In the revised version, Table 1 show the distances with the corresponding standard deviation. The relative errors are within 10% of the mean value.

Figure R7a shows the 80 individual force-distance curves and the corresponding average force-distance curve (in bold). Any individual force-distance curve shows the same features (oscillations) that the average curve. The errors of the relative positions of the layers have been determined from 10 individual force-distance curves acquired on 10 equivalent positions of the surface (Fig. R7b).

Figure R7. a. Force-distance curves. The distances are defined as in Fig. R6. b. 2D-AFM xz force maps measured on a few-layer MoS_2 surface immersed in water. The dashed lines indicate the force-distance curves used to determine the mean value and the error.

Reviewer #1 (Remarks to the Author):

Uhlig and coworkers show additional data that strongly support their conclusions. Figure R2 and its associated sentences in the rebuttal letter should be included in the Supplementary information because this provides valuable information about the origin of the force oscillation on the immersed hydrophobic surfaces and the number of hydrophobic layers. After the revision, the manuscript is suitable for publication in Nature Communications.

Reviewer #2 (Remarks to the Author):

Authors addressed all of comments carefully. I am very satisfied about the revision. The current paper can be published at is. I also recommend the paper should be considered as a VIP paper.

Reviewer #3 (Remarks to the Author):

This is a much better paper now. It is recommended to publish.

Reviewers questions are highlighted in bold.

Reviewer: 1

Uhlig and coworkers show additional data that strongly support their conclusions. Figure R2 and its associated sentences in the rebuttal letter should be included in the Supplementary information because this provides valuable information about the origin of the force oscillation on the immersed hydrophobic surfaces and the number of hydrophobic layers. After the revision, the manuscript is suitable for publication in Nature Communications.

Reply. We fully embrace the Reviewer's suggestion. The aforementioned figure appears as Supplementary Figure 9. We thank the Reviewer for suggesting this type of experiment.

Reviewer: 2

Authors addressed all of comments carefully. I am very satisfied about the revision. The current paper can be published at is. I also recommend the paper should be considered as a VIP paper.

Reply. We thank the Reviewer for his/her encouraging words and for the suggestion that the paper should be highlighted by the Editors.

Reviewer: 3

This is a much better paper now. It is recommended to publish.

Reply. We thank the Reviewer for his/her comments.